# The Effect of Tax Fairness, Peer Influence, and Moral Obligation on Sales Tax Evasion among Jordanian SMEs

Nayef Mohammad Al-Rahamneh and Zainol Bidin *

Tunku Puteri Intan Safinaz School of Accountancy (TISSA-UUM), Universiti Utara Malaysia (UUM), Sintok 06010, Kedah, Malaysia
* Correspondence: b.zainol@uum.edu.my

**Abstract:** Tax evasion remains a complex issue for tax authorities, policymakers, and researchers. While socio-psychological factors have been researched, their impact on tax evasion among SMEs has not yet been determined. This paper empirically analyses the effects of tax fairness, peer influence and moral obligation, on sales tax evasion among Jordanian SME owners/managers. A survey was used to obtain data from three regions of Jordan (north, middle, south). Random sampling was utilized in selecting the prospective respondents from SMEs in three sectors (trade, service, manufacturing). A total of 212 usable questionnaires retrieved from the SMEs were analysed using Smart-PLS 3.0. The results revealed that tax fairness and moral obligation had a significant negative effect on sales tax evasion behaviour among SME owner-managers. On the other hand, peer influence positively and significantly impacted sales tax evasion behaviour. Thus, policymakers and tax authorities should incorporate these factors in developing effective strategies to reduce tax evasion in Jordan, which could result in an improvement in the country's overall revenue collection. The findings also contribute to the scarcity of literature about the significance of socio-psychological factors in mitigating tax evasion by examining the effects of tax fairness, peer influence, and moral obligation on sales tax evasion.

**Keywords:** tax fairness; peer influence; moral obligation; tax evasion; SMEs

## 1. Introduction

Taxation is critical for both advanced and developing nations' economic growth. In underdeveloped countries, tax evasion is a major challenge for tax revenue collection (Umar et al. 2019). Governments raise tax revenue in order to provide essential social services to its citizens for the fostering of social and economic development. However, despite the varying importance of taxation for the productivity growth of both developing and developed nations, Jordan, as a developing country, confronts tax evasion challenges (Al-Rahamneh and Bidin 2022). Tax evasion is a serious phenomenon since it affects any country that relies on taxes for revenue (Islam et al. 2020; Batrancea et al. 2019). When compared with advanced economies, tax evasion activities are worsening in developing economies. For the governments, it is somewhat similar to an epidemic that they cannot control (Kassa 2021). Tax evasion affects a government's ability to improve the living standards of its citizenry and to allocate a budget for expenditure; it hinders the economic growth of countries, costing an estimated 20% of tax revenue (degl'Innocenti et al. 2022). Despite numerous attempts to address this dilemma, it remains a threatening and intractable challenge (Umar et al. 2019). Tax evasion not only depletes a nation's revenue, but also interrupts infrastructure provision, thus, harming the nation's socio-economic well-being.

Therefore, the taxpayers' ability to pay tax is crucial towards achieving a successful and long-term inflow of tax revenue. According to statistics, SMEs represent 95.1% of the companies from all businesses in Jordan, roughly 23% of the Gross Domestic Product (GDP) and 41% of total jobs (Al-Zoubi 2018; Abed 2020). SMEs play a critical role in

the economic development of a country, serving as its financial backbone. SMEs make significant contributions not only in terms of numbers, but also in the provision of jobs (Alraja et al. 2020; Talukder et al. 2020). Their contribution to income and well-being drives opportunities and business improvement, individual ability and self-confidence, political stability and social change, distributary and democratic objectives, and also reduces poverty and unemployment (Alraja et al. 2020; Sharma et al. 2020). SMEs that are sufficiently and successfully developed will have a significantly positive effect on national economic growth. According to a Jordanian study conducted by Abu Alfoul et al. (2022) from 1980 to 2018, the projected average of the hidden economy was 17.6% of the GDP, which accounts for a sizable portion. Furthermore, these findings indicated that Jordan's hidden economy grew from 11.8% in 1980 to 22.4% in 2018. Recently, Ghazo et al. (2021) stated that the approximate tax evasion in the Jordanian economy in 2016 was JOD 613.928 million. Following that, it increased to JOD 773.31 million in 2019, accounting for 2.4% of the GDP and 17 % of sales tax revenues.

Based on these statistics, there are two important aspects that need to be addressed. Firstly, there is a need to immediately act to collect sales tax revenue from the unsupervised economy. Secondly, a deeper knowledge is required to understand why taxpayers evade paying sales tax and how this problem can be addressed, or at the very least, mitigated (Alstadsæter et al. 2022). To enforce compliance, tax authorities have largely relied on deterrent measures such as conducting tax audits and investigations and imposing harsher penalties on obstinate taxpayers (Sikayu et al. 2022). Therefore, a deeper understanding of the socio-psychological factors that influence taxpayer behaviour is critical, as it supplements the limitations of deterrent measures. Socio-psychological factors such as tax fairness, peer influence, and moral obligation are not completely new in the domain of taxes. However, there is still a paucity of literature on tax evasion among SMEs. Thus, the current study provides important empirical evidence regarding the effect of tax fairness, peer influence, and moral obligation on sales tax evasion among SME owner-managers. Furthermore, tax fairness, peer influence, and moral obligation may be perceived in different ways in developing countries. Countries with well-developed socio-economic infrastructure may regard the various elements of tax fairness, peer influence, and moral obligation as insignificant in influencing the tax compliance of SME owner-managers. The contrary occurs for SMEs in developing countries, where most of the areas remain underdeveloped (Sikayu et al. 2022).

## 2. Review of Literature and Hypothesis Development

### 2.1. Tax Fairness and Tax Evasion

Tax fairness refers to the equitable payment of tax (and accompanying penalty) to tax authorities, with the incidental amount equalling the recorded amount of tax. Fairness, in its most basic form, refers to the standard of action that should really be equitable, or at the very least, acceptable and reasonable. Fairness, on the other hand, has many different aspects and, thus, many understandings of the concept (Sikayu et al. 2022). Numerous studies have discussed these aspects, including overall fairness, exchange fairness, governmental or procedural fairness, retributive fairness, vertical fairness, horizontal fairness, and individual fairness (Gilligan and Richardson 2005; Saad 2012). Tax evasion most likely occurs among taxpayers who perceive the tax system as unfair, causing them to demonstrate non-compliance to tax obligations (Sing and Bidin 2020). Tax fairness is categorised as a non-economic determinant of tax evasion and an important factor driving taxpayer behaviour (Alkhatib et al. 2019; Alm et al. 2017; Farrar et al. 2019). Kassa (2021) stated that tax fairness is described as the fairness of tax collection procedures, principles, and implementation. Due to unfairness in the tax collection procedure, immoral practices may occur. Tax fairness might positively motivate taxpayers to comply in paying tax. Taking into account the ability of SME owners-managers to pay acceptable tax rates contributes to the tax system's sustainable equity (Rantelangi and Majid 2018). Some taxpayers engage in tax evasion, while truthful taxpayers continue to meet their tax liabilities due to their belief

in the importance of doing so. Whenever the authorities raise tax rates, the additional tax might put more pressure on taxpayers, urging them to participate in tax evasion (Alsheikh et al. 2016). Ozili (2020) pointed out that the rate of tax is one factor that induces people to pay a low percentage of income taxes. Tax should be equitable and appropriate for the taxpayers. Tax fairness is an argumentative, controversial, and contentious issue, since not all taxpayers might pay the same rate of taxes (Abate 2019). Although fairness perception studies are not new in Jordan, the samples have mostly focused almost exclusively on direct taxes and their influence on tax compliance. The relationship between fairness perception among SME owner-managers and indirect tax evasion is relatively understudied, particularly in Jordan. As a result, taxpayer perceptions and their impact on tax evasion may differ from those found in other studies conducted in developed countries.

For this study, the equity theory was considered to be suitable because of its effectiveness in tackling the issue of fairness perception. The equity theory explores whether resource allocation and redistribution are equitable for all parties involved, such as tax authorities and taxpayers. Equity is evaluated by dividing the expenses of an individual's advantages and rewards. The origin of the equity theory can be traced back to Adams and Freedman (1976), a behavioural psychologist, who asserted that people seek to maintain fairness between their contributions and the rewards they expect in return, compared with the contributions and benefits of others. The equity theory postulates that people are significantly more likely to obey regulations if they have been treated equitably in a system (Bobek et al. 2013), which assumes that people perceive fairness based on how much they benefit from their contribution. The argument is that individuals desire fair and equal treatment, which determines their tendency to accept the law rules. Based on the equity theory and this correlation, it can be concluded that if taxpayers perceive inequitable treatment from the government or the appropriate tax authority, they will not comply with the tax rules, thus, resulting in a decrease in tax revenues.

Several countries have conducted studies on tax fairness focusing on SMEs. Prior studies have shown that tax fairness has a significant negative association with tax evasion (Alkhatib et al. 2019; Ariyanto et al. 2020; Jemberie 2020; Sikayu et al. 2022). Meanwhile, an unjust taxation system is related to tax evasion (Onu et al. 2019). Therefore, this current study proposed the following hypothesis based on the above arguments:

**H1.** *There is a negative relationship between tax fairness and sales tax evasion behaviour of SMEs in Jordan.*

### 2.2. Peer Influence and Tax Evasion

Peer influence is defined as pressure from friends, relatives, business colleagues, and partnerships that also has an impact on a person's decision-making (Bobek et al. 2013). In this study's context, the term 'peer' is widely used to refer to a taxpayer's peers, family, relatives, co-workers, and other acquaintances (Jackson and Milliron 1986). It is also characterised as the effect of important people on the tax behaviour decisions of SME owner-managers, contributing to the formulation of their perspectives of whether to comply or evade paying tax (Obaid et al. 2020). Chan et al. (2000), as social psychologists, provided an intellectual explanation, that when peers and close referents are evading a commitment, people close to them are more likely to act in the same manner. Furthermore, individuals may consider acting illegally once they see such violations being committed by a peer (Davis et al. 2003). Peer groups have a significant influence on a person's opinions, attitudes, and behaviour (Al Zeer et al. 2019). If taxpayers are affected by their peers in a profound manner, their decisions, personal beliefs, and attitudes would be affected as well (Puspitasari and Meiranto 2014).

Chau and Leung (2009) pointed out that taxpayers' expectations with regard to the rejection or acceptance of tax evasion, are influenced by their peers. However, from a taxation perspective, peer influence is regarded as a main factor affecting the tax behaviour of taxpayers (Alm et al. 2017). Peers have an influence on their colleagues' relevant tax

behaviour in SMEs (Maseko 2014). Hence, when a taxpayer perceives that other taxpayers are also evading tax, he or she is more likely to not file a tax return. This is in line with the social influence theory. According to Kelman (1958), the social influence theory postulates that surrounding environmental factors affect the behaviour of people, either deliberately or non-deliberately. It focuses on how other people's beliefs, views, and behaviours affect an individual (Sussman and Gifford 2013). The social influence theory is connected to the theory of social learning, which is based on Bandura's (1977) idea that an individual's personal environment has an influence on them. It also emphasises the effect of peer opinions and level of social influence on an individual's relationships in socialisation, as being essential in identifying tax evasion behaviour (Sutinen and Kuperan 1999).

Although peer influence studies are not new in Jordan, the majority have largely focused on direct taxes, with only a few focusing on indirect taxes and their impact on tax compliance. The relationship between peer influence and indirect tax evasion among SME owner-managers is generally understudied, notably in Jordan. Therefore, taxpayer perception and its effects on tax evasion may differ from those in other studies on developed countries.

Based on the social influence theory and this relationship, it can be concluded that taxpayers' decisions of whether to comply with, or evade, tax payments, are influenced by their peers' opinions. Thus, having a more negative perception of others' tax evasion behaviour may increase tax compliance, and vice-versa (Çevik and Yeniçeri 2013). Some prior studies found that peer influence had a significant positive effect on tax evasion (Abdixhiku et al. 2018; Bhutta et al. 2019; Bidin and Sinnasamy 2018). In areas other than taxes, i.e., zakat, also shows that peers influence the behavior of compliance with zakat payment (Bidin et al. 2009). Therefore, this study contributes to bridging the gap. The following hypothesis was developed based on the above discussion:

**H2.** *There is a positive relationship between peer influence and sales tax evasion behaviour of SMEs in Jordan.*

### 2.3. Moral Obligation and Tax Evasion

Moral obligation is the internalisation of positive behaviour norms by a person, which is mostly affected by internal constraints (Sabucedo et al. 2018). Milesi and Alberici (2018) stated that an individual's moral obligation should encourage them to comply with their conscience, despite what it may cost and whether it is likely to succeed. Moral obligation refers to an individual's internal motivation for engaging in ethical norms of behaviour, which primarily impacts human behaviour through internal restrictions (Sabucedo et al. 2018). From the perspective of tax, moral behaviour refers to the internal motivation or desire to pay tax (Hosseini Kondelaji et al. 2016). This in line with the assertion of Young et al. (2016) who defined moral behaviour as an intrinsic motivation to comply with ethical values and moral principles.

Moral obligation is frequently specified as essential for developing a better understanding of tax compliance among taxpayers. Moral obligation is a factor that can discourage taxpayers from engaging in tax evasion. It is the responsibility of taxpayers to pay an appropriate taxable amount to the tax department without relying on third-party enforcement. It is basically a source of intrinsic motivation for people who pay tax (Sadjiarto et al. 2020). The moral obligation to pay taxes is based on ethics, or the perception of what is right or wrong (Alm and Torgler 2011). When taxpayers demonstrate low tax morality, they are less likely to pay their due taxes and are more likely to engage in tax evasion (Oberholzer 2008; Torgler 2006; Torgler et al. 2008). According to Frey and Torgler (2007), tax morality or taxpayer sincerity will improve when tax officials are respectful and considerate in their responsibilities toward them.

According to Ozili (2020), there are three main perspectives on the morals and ethics of tax evasion. The first argument is that tax evasion is immoral and, therefore, no taxpayer should engage in it. The second argument states that the authority is unlawful and has

no ethical authority to take things from taxpayers. The final viewpoint is that tax evasion might be permissible in some instances but not in others; as a consequence, the willingness to engage in tax evasion is an ethical dilemma that includes a variety of factors (McGee et al. 2012). The moral obligation of taxpayers determines whether or not they engage in tax evasion. It can influence taxpayers' willingness to engage in tax evasion (Nangih and Dick 2018). Taxes levied by the appropriate authority are known to be ethical, which is consistent with the moral development theory.

Kohlberg (1969), who extended Piaget's theory, stated that moral development is a continuous lifelong mechanism. The moral development theory postulates that attaining a higher moral obligation will motivate individuals to comply with rules. In the taxation context, Alm and Torgler (2011) stated that moral obligation for tax law is based on ethics, i.e., what the society perceives as right or wrong. In connection with tax evasion, the moral development theory indicates that taxpayers with high moral obligation will sacrifice their own benefits for the sake of others. It also emphasises the effect of moral obligation and its level on the taxpayer's relationships as the main factor determining evasion (Kohlberg and Hersh 1977). Consequently, there is a need to determine the impact of moral obligation on sales tax evasion behaviour. Moral obligation is integrated in this study because it reflects substantial encouragement to execute a particular behaviour. Prior studies have revealed a negative and significant influence of moral obligation on tax evasion (Alleyne and Harris 2017; Braithwaite et al. 2010; Culiberg 2018; Owusu et al. 2019). The current study proposed the following hypothesis based on the foregoing discussion:

**H3.** *There is a negative relationship between moral obligation and sales tax evasion behaviour of SMEs in Jordan.*

### 3. Theoretical Framework

The theoretical framework of this study was established and supported by the integration of social psychological theories, which imply that tax fairness, peer influence, and moral obligation impact tax evasion behaviour. The relationship between the study's independent variables and sales tax evasion among SMEs was explained using these theories. The equity theory postulates that people are significantly more willing to comply with laws if they have been treated fairly in the system. Hence, it can be assumed that people perceive fairness based on how much they benefit from their contribution. However, the social influence theory focuses on how other people influence an individual's feelings, opinions, or conduct, and how an individual is impacted by their surroundings. Furthermore, the moral development theory proposes that achieving a higher moral obligation will inspire people to follow norms. Based on these theories, the above hypothesis, and the findings of previous studies, the theoretical framework for the present study is proposed as illustrated in Figure 1.

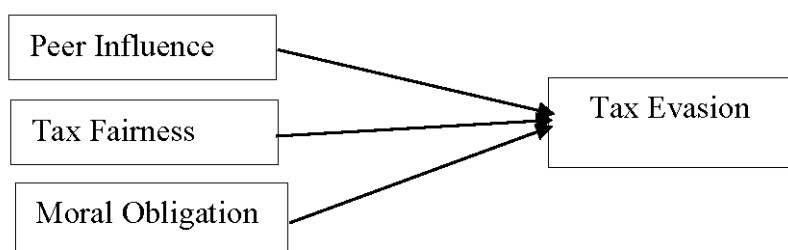

**Figure 1.** Theoretical Framework.

### 4. Research Methodology

This study employed a quantitative research design in determining the effect of tax fairness, peer influence, and moral obligation on tax evasion in Jordan. Furthermore, when examining the relationships between variables and testing hypotheses, quantitative design methods are the most appropriate and logical. The common objective of quantitative

research is to test theories or answer specific research questions (Zikmund et al. 2013). In both developing and developed countries, studies of tax behaviour generally utilise quantitative methodologies (Fjeldstad and Semboja 2001; Saad 2011; Verboon and van Dijke 2011). Creswell and Creswell (2017) further affirmed that a quantitative approach was appropriate for complex studies involving numerous factors. Predetermined instruments and closed-ended questions can be used to examine the variables in consideration, permitting statistics and data to be analysed using statistical processes (Creswell and Creswell 2017; Trochim and Donnelly 2008).

The target population for this study entailed Jordanian SME owner-managers, whose perspectives are important in understanding the factors influencing sales tax evasion behaviour. There are an estimated 166,154 SMEs in Jordan (Jordanian Department of Statistics 2021) from which a sample size of 382 was drawn. It was measured by employing Krejcie and Morgan's (1970) criterion. However, Israel (1992) recommended increasing the sample size by at least 30% to offset potential non-responses. Thus, the sample size was expanded to 500 to obtain more respondents and to tackle the probability of non-responses. A total of 500 questionnaires were distributed directly to the target population, i.e., the SME owner-managers using mail, online and drop-off modes.

The present study obtained 212 completed questionnaires, representing a 42.4% response rate which was considered as adequate and reasonable. All questionnaire items were measured using a five-point Likert scale whereby (1) denoted strongly disagree, and (5) denoted strongly agree. The 14 items for sales tax evasion were adapted from Gilligan and Richardson (2005). Another seven items were used to measure tax fairness, which were all adapted from Gilligan and Richardson (2005). Six items were adapted from Bobek et al. (2007) to measure peer influence. In addition, four items measured moral obligation. One item was adapted from Bobek and Hatfield (2003), and another three were adapted from Beck and Ajzen (1991).

## 5. Data Analysis and Results

The data in the current study was analysed using the partial least squares structural equation modelling (PLS-SEM) analysis technique using Smart PLS 3.3.9. PLS-SEM is a statistical technique that is gaining the attention of many researchers for analysing empirical data in a variety of areas, including tax behaviour (Farouk et al. 2018). PLS-SEM modelling is suitable for analysing complicated models with a large number of items, variables, and relationships (Chin 2010). In this study, PLS-SEM was used for evaluating the measurement and structural models. PLS-SEM comprises a two-stage analytical technique as recommended by Hair et al. (2021), entailing the measurement model (indicator reliability, internal consistency reliability, convergent validity, and discriminant validity) and the structural model, namely, testing the path coefficients β significance through bootstrapping procedures for a direct association, coefficient of determination ($R^2$), effect size ($f^2$) for a direct association, and determining the predictive relevance of the model (Q2).

### 5.1. Assessment of Measurement Model

Convergent validity and discriminant validity were the two types of models examined to evaluate the reflective measurement model prior to hypothesis testing. The indicators loading, average variance extracted (AVE), Cronbach's Alpha (CA), and composite reliability (CR) are all used to evaluate the measurement's convergent validity. On each construct, the following results were found following the measurement assessment: (i) as suggested by Hair et al. (2011), the indicators' loadings were found to be higher than 0.60, thus, achieving the cut-off point of 0.4 to 0.7; (ii) the AVE values were also higher than the minimum criterion of 0.50, i.e., ranging from 0.516 to 0.787; (iii) the CA and CR values ranged from 0.766 to 0.965, and 0.841 to 0.969, respectively, thus, meeting the acceptable value of 0.70 for CR and CA evaluations (Hair et al. 2017; Salleh et al. 2016). Consequently, the present study was considered to have acceptable convergent validity based on the findings. Table 1 and Figure 2 summarise the results of convergent validity.

**Table 1.** Convergent validity for reflective measurement model of the constructs.

| Construct | Items | Loading | CA | CR | AVE |
|---|---|---|---|---|---|
| Tax Fairness | TF1 | 0.711 | 0.910 | 0.937 | 0.787 |
| | TF2 | 0.729 | | | |
| | TF3 | 0.749 | | | |
| | TF4 | 0.844 | | | |
| | TF5 | 0.803 | | | |
| | TF6 | 0.802 | | | |
| | TF7 | 0.699 | | | |
| Peer Influence | PI1 | 0.731 | 0.766 | 0.841 | 0.516 |
| | PI2 | 0.762 | | | |
| | PI3 | 0.737 | | | |
| | PI4 | 0.731 | | | |
| | PI5 | 0.622 | | | |
| Moral Obligation | MO1 | 0.832 | 0.965 | 0.969 | 0.690 |
| | MO2 | 0.887 | | | |
| | MO3 | 0.912 | | | |
| | MO4 | 0.915 | | | |
| Tax Evasion | TE1 | 0.758 | 0.882 | 0.907 | 0.584 |
| | TE2 | 0.802 | | | |
| | TE3 | 0.918 | | | |
| | TE4 | 0.896 | | | |
| | TE5 | 0.905 | | | |
| | TE6 | 0.785 | | | |
| | TE7 | 0.878 | | | |
| | TE8 | 0.869 | | | |
| | TE9 | 0.628 | | | |
| | TE10 | 0.787 | | | |
| | TE11 | 0.864 | | | |
| | TE12 | 0.848 | | | |
| | TE13 | 0.782 | | | |
| | TE14 | 0.866 | | | |

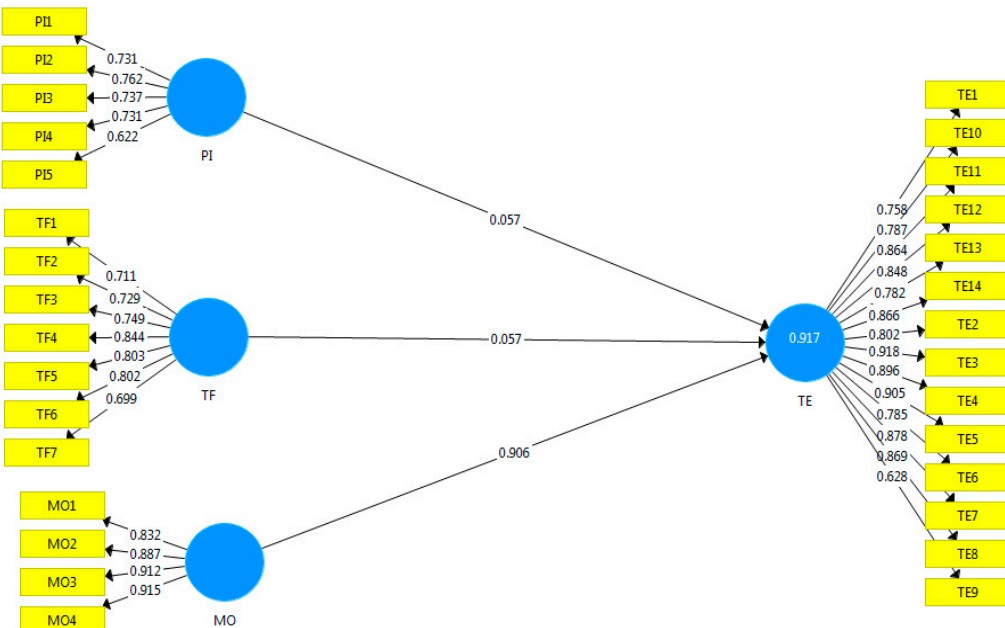

**Figure 2.** The measurement model.

Discriminant validity was mainly used to determine the construct validity of the measurement models (Hair et al. 2017). Fornell and Larcker's (1981) criteria and heterotrait–

monotrait (HTMT) ratio were utilised to assess the discriminant validity. As suggested by Hair et al. (2017) the square root of the AVE of each construct should be higher than its highest correlation with any other construct in the model. The square root of AVE ranged between 0.718 to 0.887, which was greater than any of the other latent variables' correlation. These results indicated that the discriminant validity was sufficient for further research. The bolded values on the diagonals represent the square root of the AVE, which were greater than the corresponding row and column values. Fornell and Larcker's (1981) criteria are presented in Table 2 below.

**Table 2.** Fornell and Larcker's (1981) criteria.

|  | **MO** | **PI** | **TE** | **TF** |
|---|---|---|---|---|
| Moral Obligation | **0.887** | | | |
| Peer Influence | 0.389 | **0.718** | | |
| Tax Evasion | 0.459 | 0.424 | **0.831** | |
| Tax Fairness | 0.462 | 0.245 | 0.489 | **0.764** |

The heterotrait–monotrait (HTMT) ratio of correlations was used to assess the discriminant validity, which shows the extent to which constructs are distinctive from one another (Henseler et al. 2015). Henseler et al. (2015) and Kline (2015) recommended a threshold of 0.85, for the constructs to be determined as conceptually different. Table 3 includes a summary of the discriminant validity result, which was demonstrated by values of less than 0.85.

**Table 3.** HTMT ratio of the constructs (N = 212).

| **Construct** | **MO** | **PI** | **TE** | **TF** |
|---|---|---|---|---|
| Moral Obligation | - | | | |
| Peer Influence | 0.446 | - | | |
| Tax Evasion | 0.488 | 0.478 | - | |
| Tax Fairness | 0.495 | 0.287 | 0.509 | - |

The measurement model constructed utilising Smart PLS is shown in Figure 2 below.

### 5.2. Assessment of the Structural Model

Using a 5000-bootstrapped sample, the significance of the path coefficients was assessed using t-statistics and *p*-values derived from the structural model of PLS (Hair et al. 2017). The statistical estimates of the structural model path coefficients are presented in Table 4 and Figure 3 below. The first hypothesis predicted a negative relationship between tax fairness and sales tax evasion based on the hypothesis development; however, the findings showed a significant relationship ($\beta = -0.149$, t = 2.823, $p < 0.05$), which assumed that an increase in tax fairness would lead to a decrease in tax evasion. It also indicated that the relationship was statistically significant. This meant that there was enough evidence to support the established relationship between tax fairness and tax evasion. The *p*-value was 0.003. Therefore, hypothesis 1 was supported.

**Table 4.** Path coefficients of the exogenous constructs predicting tax evasion.

| Hypotheses: Path | Path Coefficients (β) | SD | *t*-Value | *p*-Value | Decision |
|---|---|---|---|---|---|
| H1: Peer Influence → Tax Evasion | 0.057 | 0.083 | 1.687 | 0.011 ** | Supported |
| H3: Tax Fairness → Tax Evasion | 0.057 | 0.025 | 2.330 | 0.003 ** | Supported |
| H15: Moral Obligation → Tax Evasion | 0.906 | 0.020 | 2.229 | 0.000 ** | Supported |

** $p \leq 0.05$, t = 1.645.

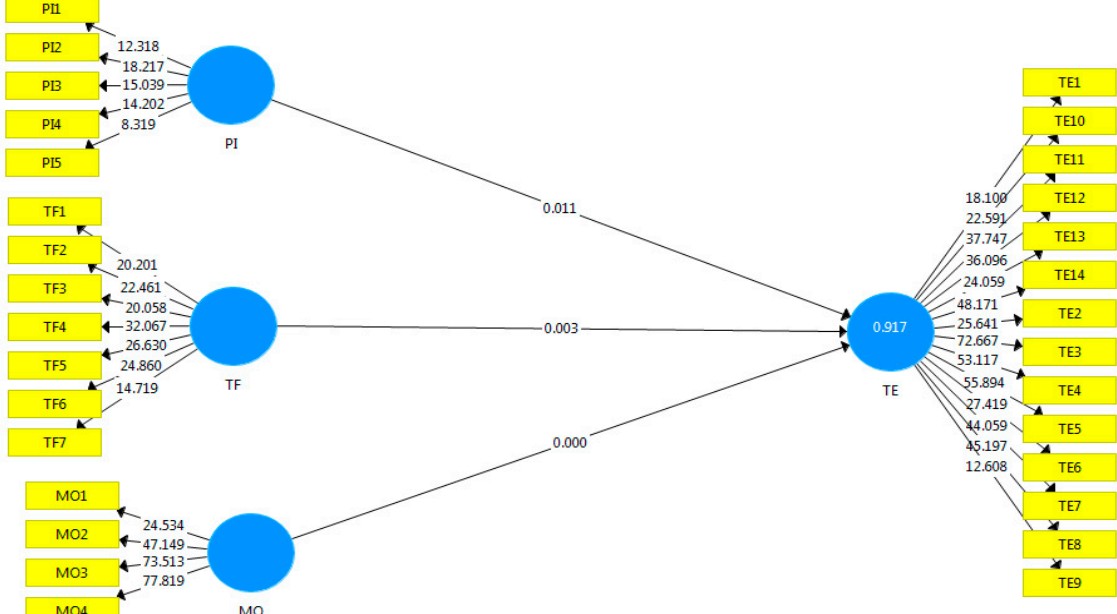

**Figure 3.** Structural model for direct effect (*p*-values).

The second hypothesis postulated a positive relationship between peer influence and sales tax evasion. The findings showed a positive relationship (β = 0.057, t = 1.687, $p < 0.05$), which also signified that an increase in peer influence would result in an increase in tax evasion. The t-statistic was 1.687, indicating statistical significance in the relationship. The P-value was 0.011 which also signified that there was sufficient evidence to support the second hypothesis. The third hypothesis proposed a negative relationship between moral obligation and sales tax evasion. The findings showed a negative relationship (β = −0.906, t = 2.229, $p < 0.05$), which presumed that an increase in moral obligation would reduce the prevalence of tax evasion. It also indicated that the relationship was significant statistically. This illustrated that there was adequate evidence to confirm the established relationship between moral obligation and sales tax evasion. The *p*-value was 0.000 which indicated that there was sufficient evidence to support the third hypothesis.

The coefficient of determination ($R^2$) is one of the most important criteria for evaluating structural models using PLS-SEM. In a research model, the $R^2$ explains how one or more exogenous latent constructs explain the variation in the endogenous latent construct (Hair et al. 2017). The $R^2$ value for the current study was 0.917 as shown in Figure 3 and Table 5, indicating that the exogenous latent variables together explained 91.7% of the variance in tax evasion. Meanwhile, other factors outside the current model explain the remaining percentage.

**Table 5.** Coefficient of determination, $R^2$.

| | $R^2$ | $R^2$ Adjusted |
|---|---|---|
| **Tax Evasion** | 0.917 | 0.509 |

The effect size ($f^2$) was the criterion used to evaluate the PLS structural model. After calculating the $R^2$, the $f^2$ was evaluated to determine if an exogenous variable's influence on the endogenous variable was significant (Hair et al. 2017). In a structural model, the $f^2$ is the change in $R^2$ in the endogenous construct when the exogenous construct is either retained or omitted. Cohen (1988) classified $f^2$ into three groups in the structural model: 0.02 to 0.14 denotes a small effect, 0.15 to 0.34 denotes a medium effect, and 0.35 and above denotes a large effect. Based on Table 6, the $f^2$ values for moral obligation, peer influence, and tax fairness were 0.001, 0.033, and 0.031, respectively. The direct relationship's effect size on tax evasion ranged from no effect to a small effect. This means that the missing construct in the model had no effect on the endogenous construct. It was, however, at an acceptable level.

**Table 6.** Effect sizes, $f^2$.

| Construct | $f^2$ | Effect Size |
|---|---|---|
| Moral Obligation | 0.001 | No effect |
| Peer Influence | 0.033 | Small Effect |
| Tax Fairness | 0.031 | Small Effect |

Predictive relevance ($Q^2$), a resampling technique, is another important statistics value (Hair et al. 2017). The blindfolding technique was used with single or multiple items in a reflective measurement model to assess the predictive relevance of the endogenous variable (Hair et al. 2017). As recommended by Hair et al. (2017) and Henseler et al. (2009), a cross-validated redundancy measure ($Q^2$) value higher than zero indicated the model's predictive importance. The results of this test revealed that the study's model had predictive relevance value for the endogenous variable, as illustrated in Table 7.

**Table 7.** Predictive relevance ($Q^2$).

| Endogenous Construct | SSO | SSE | $Q^2$ (=1-SSE/SSO) |
|---|---|---|---|
| Tax Evasion | 2968.00 | 1117.688 | 0.623 |

## 6. Discussion

This study aimed to examine the socio-psychological factors affecting sales tax evasion among Jordanian SMEs. Tax fairness was found to be negatively and significantly related to sales tax evasion for the direct relationship hypothesis. Hence, this hypothesis was supported, suggesting that an increase in tax fairness perception will result in a decrease in sales tax evasion. The result was consistent with those of Alkhatib et al. (2019), Ariyanto et al. (2020), Jemberie (2020), and Sikayu et al. (2022), which showed a negative and significant relationship between tax fairness and tax evasion. It can be concluded that in the Jordanian context, tax fairness is regarded as an important determinant that impacts tax evasion. Specifically, if taxpayers believe the government or the appropriate tax authority has treated them fairly, they will willingly comply with the tax rules, resulting in an increase in tax collections. Next, this study found that peer influence was positively related to tax evasion. This result was congruent with that of Abdixhiku et al. (2018), Bhutta et al. (2019), Bidin et al. (2009) and Bidin and Sinnasamy (2018), which found a positive relationship between peer influence and tax evasion. This may be because peer influence is defined as one of the social influence factors that determine tax evasion behaviour. According to common thinking, taxpayers' behaviours are affected by their peers' opinions about

whether they are willing to evade or pay tax imposed by the government or the tax authority, and whether they would wilfully violate tax rules, resulting in a reduction in tax collections. As a result, holding a negative opinion about the tax evasion behaviour of other society members may minimise tax compliance, and vice versa. Therefore, taxpayers are more willing to engage in tax evasion behaviour if their peers engage in tax evasive behaviour, in order to avoid financial penalties. The hypothesis of moral obligation was proven to be supported. This result was consistent with the theory of those who believe that moral obligation is an important determinant of tax evasion behaviour (Alleyne and Harris 2017; Braithwaite et al. 2010; Culiberg 2018; Owusu et al. 2019). It is important to note that moral obligation is contingent on one's ethical beliefs, which are convictions about what is right and wrong (Alm and Torgler 2011). The moral obligation for paying tax is based on ethics, which society considers to be wrong or right. Clearly, moral obligation as a positive internal value may have a greater influence on tax evasion behaviour.

## 7. Conclusions, Policy Implications, Limitations and Suggestions for Further Studies

Tax evasion among small and medium enterprises (SMEs) is a global challenge. Since SMEs constitute the largest segment of the business sphere, tax evasion by SMEs has garnered considerable attention. Hitherto, various psychological aspects related to tax evasion have been studied in the past, but no conclusive results have been established. Despite the design and implementation of numerous strategies, tax evasion continues to be a problem. Various aspects of evasion could be addressed to ensure optimal compliance. This study integrated social and psychological determinants of sales tax evasion to provide empirical evidence, and, moreover, to provide a novel viewpoint on the issue of tax evasion that may assist better understanding of this problem. It also offers additional insight into the relationship between SME owner-managers tax-paying determinants and tax evasion.

This study examined the effect of tax fairness, peer influence, and moral obligation on the behaviour of SME owner-managers towards tax evasion. As per the findings, tax fairness and moral obligation have a significant and negative impact on taxpayers' sales tax evasion behaviour. On the other hand, peer influence has a significant positive effect on sales tax evasion. These results indicate that taxpayers' perspectives on tax evasion are affected by their perceptions of fairness, peer influences, and moral obligation. This suggests that sales taxpayers may be surrounded by evading peers, with their compliance behaviour being positively related to high peer influence.

Moreover, the finding provides tax authorities with a better understanding of how to design and develop new strategies to combat tax evasion. In addition, the results highly recommended tax administrations, not only in Jordan but also in other Middle East countries, to develop policies based on the determinants of tax evasion behaviour. There is a significant budget deficit in Jordan. Large-scale tax evasion contributes to Jordan's enormous budget deficit. The budget deficit will be covered and eradicated if the government is able to defeat, curtail, or alleviate tax evasion. As a result, the government will find support if the issue of tax evasion is resolved, and this support will allow the government and tax authorities close the deficit that endangers the state's economy. By considering the context of other developing countries, particularly for Middle Eastern countries where there is little research on sales tax evasion, the current study provides empirical support for the crucial factors affecting tax evasion. These findings concerning the causes of tax evasion may also be extended to sales tax evasion activities in other developing nations that have similar business conditions, policy settings, and characteristics of their taxpayers.

This study, as with any other research, has its limitations. Although the current research makes important theoretical and practical contributions, there are still some important limitations that need to be acknowledged. Future research opportunities are created by these limitations. This study did not cover all conceivable factors that influence a taxpayer's decision to pay sales tax. Self-reported surveys, as well as other compliance studies, might not fully reflect respondents' real behaviour (Van Dijke and Verboon 2010).

This limitation is especially present when information is needed for sensitive tax-related embarrassments. Another limitation was that the present study's response rate was just 42.4%, despite the numerous follow-ups conducted following the survey distribution. However, for Jordanian SME owners and managers, this was acceptable. Data collection was hampered by the respondents' unwillingness to comply. The sensitive nature of tax-related issues, particularly tax evasion, and a lack of enthusiasm to fill out the questionnaire can be attributed to this.

Hence, future studies can extend the research framework to include more socio-phycological, economic, and behavioural factors, enabling the findings to be generalized. Additionally, longitudinal research might be conducted to determine if the results shift over time. Last, but not least, future studies should investigate the indirect impact of the determinant factors on tax evasion by exploring the mediating or moderating role of moral obligation, trust, and ethics.

**Author Contributions:** The research was done independently. The authors read and approved the final manuscript. All authors have equally contributed to all phases and parts of the manuscript. All authors have read and agreed to the published version of the manuscript.

**Funding:** The authors received no financial support for the research, authorship and/or publication of this article.

**Institutional Review Board Statement:** Not applicable.

**Informed Consent Statement:** Not applicable.

**Data Availability Statement:** Available upon request.

**Conflicts of Interest:** The authors declare that the research was conducted in the absence of any commercial or financial relationships that could be construed as a potential conflict of interest.

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
