# Peer review of "The Effect of Tax Fairness, Peer Influence, and Moral Obligation on Sales Tax Evasion among Jordanian SMEs"

_jrfm, doi:10.3390/jrfm15090407_

Round 1

Reviewer 1 Report

I think that the paper is an interesting piece of research in the field of tax morale related to tax evasion. The survey and the results are well discussed. I only suggest to the Authors a more deep discussion about the policy implications in section 7. Furthermore, a brief summary in the abstract of the main findings and the policy implications for the SMEs would be helpful for the reader.

Reviewer 2 Report

Dear authors,

Please find my observations below:

Please use the standard template of the journal.

In formulating the hypotheses, I would suggest to remove the article “the”. For instance, “There is a negative relationship between tax fairness and sales tax behaviour of SMEs in Jordan”.

Format references in a uniform manner.

In the last section, please mention the study limitations.

The reference list could be increased with the following article:

         Batrancea, L., Nichita, A., Olsen, J., Kogler, K., Kirchler, E., Hoelzl, E., …, Zukauskas, S., 2019. Trust and power as determinants of tax compliance across 44 nations. J. Econ. Psychol. 74, 102191. https://doi.org/10.1016/j.joep.2019.102191.

Reviewer 3 Report

The article discusses an interesting topic which is the influence of tax fairness and moral obligation on sales tax evasion. The article indicates a very good knowledge of the subject. An in-depth review of the literature, which points to the latest publications on the analyzed topic, deserves praise. The hypotheses are clear, their verification does not raise any objections. The response rate is impressive. 

The only doubt concerns the veracity of the answers given by Jordanian SME owner-managers in the item of moral obligation. It will be worth stressing this limitation.

Reviewer 4 Report

Thank you for the opportunity to read this research paper. Positively evaluating the research I consider it necessary to raise certain debatable issues that require consideration. Here are some comments that I hope will be helpful to improve performance. 
